# Electromagnetic Wave Absorption and Mechanical Properties of CNTs@GN@Fe_3_O_4_/PU Multilayer Composite Foam

**DOI:** 10.3390/ma14237244

**Published:** 2021-11-27

**Authors:** Chunfu Gao, Xinsheng He, Fengchao Ye, Shuxin Wang, Guang Zhang

**Affiliations:** 1Key Laboratory of Urban Rail Transit Intelligent Operation and Maintenance Technology, Equipment of Zhejiang Province, College of Engineering, Zhejiang Normal University, Jinhua 321004, China; cfgao2007@zjnu.cn (C.G.); xsh@zjnu.cn (X.H.); zhongjiaqi.love@163.com (F.Y.); 2School of Intelligent Manufacturing and Electronic Engineering, Wenzhou University of Technology, Wenzhou 325035, China; 3College of Mechanical Engineering, Zhejiang University of Technology, Hangzhou 310023, China; guangzhang@zjut.edu.cn

**Keywords:** multilayer magnetic foam, CNTs@GN@Fe_3_O_4_/PU, absorbing performance, mechanical behavior

## Abstract

With the development of intelligent communications and stealth technology in the military field, electromagnetic wave pollution cannot be ignored, and absorbing materials have entered people’s field of vision and gradually become a research hotspot. The ideal absorbing material should have the characteristics of “strong, wide, thin, and light”, but a single absorbing material often cannot meet the above conditions. At present, absorbing metal powder combined with two-dimensional carbon nanomaterials (such as carbon nanotubes, graphene, etc.) has became a trend. This article focus on a three-layer composite of Fe_3_O_4_, Carbon nanotubes@ Fe_3_O_4_, Carbon nanotubes@Graphene nano-platelets@ Fe_3_O_4_, which was synthesized by solvothermal method. The results show that the electromagnetic wave absorption performance of the three-layer foam at a thickness of 3.0 mm is more excellent. The minimum of RL can reach −67.0 dB, and the effective bandwidth is above 5.0 GHz. All this is due to the synergy of dielectric and magnetic loss between Fe_3_O_4_, CNTs, and GN, the increase of interface polarization and the path of electromagnetic wave reflection and scattering by three-layer foam.

## 1. Introduction

Currently, with the widespread use of radio technology, electromagnetic wave radiation has caused serious environmental pollution and endangered human health [1,2,3,4,5]. In order to eliminate electromagnetic wave pollution, people have continually been developing new and efficient electromagnetic wave absorbing materials (EMWA). Ordinary metal-based EMWA materials have disadvantages such as poor corrosion resistance, high density, and poor electromagnetic wave absorption and adjustment capabilities [6,7], and have difficulties in meeting the current demand for flexible electronic products. Therefore, flexible materials with the characteristics of “strong, wide, thin, and light” have gradually attracted people’s attention and research.

Carbon-based materials such as carbon nanotubes, graphene, etc., are widely used as electromagnetic wave absorbers. Carbon-based electromagnetic wave absorbing materials (EWM) exhibit high attenuation capabilities by effectively converting electromagnetic waves into heat, with low cost and high chemical stability, low density, and other advantages [8,9,10]. However, the high conductivity of carbon materials will lead to weak impedance matching, which limits its microwave absorption performance [11,12].

As an absorbent, Fe_3_O_4_ ferrite is widely used due to its excellent magnetic loss performance. Fe_3_O_4_ is an inverse spinel type ferrimagnetic material of Fe^2+^ and Fe^3+^ with different valences. Disordered ions are distributed in an octahedral structure, and free electrons can be quickly exchanged between iron ions, resulting in excellent magnetic loss performance [13,14]. In order to further improve the performance of EMWA materials, a new type of wave absorbing agent has been prepared by compounding carbon materials and Fe_3_O_4_ through many methods. He et al. [15] coated the surface of hollow fly ash microspheres (HFAM) with a nano-Fe_3_O_4_ layer to form a HFAM/Fe_3_O_4_ absorbent. The cement-based material functionalized with HFAM/Fe_3_O_4_ shows excellent Electromagnetic wave absorption performance, and its bandwidth is close to 8 GHz when the bandwidth is lower than −10 dB, which indicate that this new HFAM/Fe_3_O_4_ absorbent modified cement-based material has potential. Li et al. [16] used a simple solvothermal method to prepare nano-Fe_3_O_4_ compact-coated CNTs (FCCs) and Fe_3_O_4_ loose-coated CNTs (FLC). The results attest that the difference in microwave absorption between FLC and FCC is due to the difference in complementarity between dielectric loss and magnetic loss, which is related to the coverage density of Fe_3_O_4_ nanoparticles on the surface of carbon nanotubes. For FCC, the mass ratio of CNTs to Fe^3+^ is then optimized to maximize the effective complementarity between dielectric loss and magnetic loss. The FCC under the optimized ratio of CNTs to Fe^3+^ shows the most effective ratio of RL_min_ (−28.7 dB) and an effective bandwidth of 8.3 GHz. Song used simple hydrothermal method to prepare MWCNT/Fe_3_O_4_ composites. Ferromagnetic matrix composites with light weight, controllable morphology, and strong microwave absorption ability were prepared by adjusting particle size and holding time. Through the selection of functional groups and solutions, the directional agglomeration of Fe_3_O_4_ is realized to form a complex network structure. Finally, when the solvothermal time is 15 h and the size of microspheres is 400 nm, the reflection loss is −38 dB

Because a single absorbing coating often has a relatively narrow effective absorption bandwidth, it cannot meet the actual needs well [17,18]. For the design of multi-layer absorbing coatings, many factors need to be considered, such as the type of material, the thickness of the coating, the proportion and distribution of components in a single layer, etc. [19]. Therefore, this article proposes a method to prepare nano Fe_3_O_4_/CNTs@Fe_3_O_4_/CNTs @GN@ Fe_3_O_4_ composite particles by solvothermal method. The three powder particles are dispersed in a single-component moisture-curable polyurethane, and a three-layer magnetic foam is prepared by the layer-by-layer method to achieve the purpose of enhancing absorption and broadening the frequency band.

## 2. Experimental Section

### 2.1. Materials

Polyether polyurethane (PU) was supplied by Jining Huakai Resin Co., Ltd. (Shandong, China). Carbon nanotubes (CNTs) with 9 nm outer diameter and 1.5 μm length on average were provided by Nanocyl SA. Iron(III) Chloride Hexahydrate (FeCl_3_·6H_2_O), Sodium Citrate Dihydrate (Na_3_C_6_H_5_O_7_.2H_2_O), Sodium Acetate (CH_3_COONa), Nitric Acid (HNO_3_), Absolute Ethyl Alcohol, and Ethylene glycol (EG) were purchased from Sinopharm Chemical Reagent Co., Ltd. (Shanghai, China). Graphene nano-platelets (GNPs), with an average diameter of 10 μm and a thickness of about 10 nm, were purchased from Wuxi CAIDA Materials Technology Co., Ltd., Wuxi, China.

All reagents were of analytical grade and used as received without further purification.

### 2.2. Synthesis of Fe_3_O_4_/CNTs@Fe_3_O_4_/CNTs@GN@Fe_3_O_4_ Magnetic Nanoparticles

The specific preparation process is shown in Figure 1. Firstly, weigh 50 mg of CNTs and GN and ultrasonically disperse them in a 68% concentration of nitric acid solution, fully disperse for 1 h, and then transfer them to a high temperature reactor lined with PTFE. Heat at 140 °C for 8 h, until the reaction kettle is cooled, take out the reaction product, and wash it repeatedly with absolute ethanol and deionized water until pH = 7. Place the cleaned product in a vacuum drying oven at 60 °C for 12 h and vacuum dry, and finally the dried product can be ground for use. Functional groups such as hydroxyl and carboxyl groups will be generated on the surface of acidified CNTs and GN to facilitate the subsequent adhesion and growth of Fe_3_O_4_.

Then, solvothermal method was used to further synthesize CNTs@Fe_3_O_4_/CNTs @GN@Fe_3_O_4_ magnetic nanoparticles. For CNTs@ Fe_3_O_4_, firstly dissolve 3.258 g FeCl_3_·6H_2_O in 60 mL ethylene glycol using magnetic stirring. Subsequently, add 10.8 g of CH_3_COONa and 0.751 g of Na_3_C_6_H_5_O_7_·2H_2_O and stir fully to obtain a yellow solution. Then, weigh 50 mg of acidified CNTs in 10 mL of absolute ethanol, and place the suspension in an ultrasonic vibrator to fully shake for 30 min. After shaking, add the ethanol mixed suspension of CNTs to the previous yellow solution and continue magnetic stirring for 1 h. Finally, transfer the resulting mixture to a 100 mL polytetrafluoroethylene lined stainless steel reactor and heat at 200 °C for 10 h. After the heating time is over, allow the reaction kettle to cool naturally, and wash the black product obtained from the reaction repeatedly with deionized water and absolute ethanol more than 5 times. Dry the cleaned product in a vacuum drying oven at 60 °C for 8 h under the protection of 5 mL of absolute ethanol. Finally, CNTs@Fe_3_O_4_ magnetic nanoparticles are obtained, and the addition amount of CNTs is set at 50 mg and 70 mg, so the sample particles are named 50 CNTs@Fe_3_O_4_ and 70 CNTs@Fe_3_O_4_.

For the preparation of CNTs@GN@Fe_3_O_4_, repeat the above steps. At the same time change the ratio of CNTs to GN. The initial addition amount is still 50 mg and 70 mg, two specifications. Make the mass ratio of 1:1 (C1G1), 1:2 (C1G2) and 2:1 (C2G1) to prepare six kinds of powder samples, named 50C1G1, 50C1G2, 50C2G1, 70C1G1, 70C1G2, and 70C2G1.

### 2.3. Preparation of the Three-Layer Composite

Through the layered pouring method of LBL (layer by layer), a multi-layer composite material is produced, which is shown in Figure 2. The overall thickness of the material is controlled at 3.0 mm by the mold, and the thickness of each layer is 1.0 mm. Fe_3_O_4_/PU is used as the matching layer of the first layer. Since Fe_3_O_4_ has a low dielectric constant, it can be used as a wave-transmitting layer. The second layer is the loss layer, with CNTs@Fe_3_O_4_/PU as the second layer. Taking CNTs@GN@Fe_3_O_4_/PU as the third layer, this layer functions as the lossy layer and the reflective layer as a whole. The mass ratio of each layer of the absorber above is 15 wt%. Six specifically designed structural combinations are used, Fe_3_O_4_-50CNTs@Fe_3_O_4_-50C1G2, Fe_3_O_4_-50CNTs@Fe_3_O_4_-50C2G1, Fe_3_O_4_-50CNTs@Fe_3_O_4_-70C1G1, Fe_3_O_4_-70CNTs@Fe_3_O_4_-50C1G2, Fe_3_O_4_-70CNTs@Fe_3_O_4_-50C2G1 and Fe_3_O_4_-70CNTs@Fe_3_O_4_-70C1G1, which are named S1, S2, S3, S4, S5, and S6, respectively.

Use an automatic stirrer to stir for 30 min at a speed of 600 r/min. Then, pour the obtained mixture into a mold, put the mold into a vacuum box to extract the air in the mixture. Then, put the mold into an ultrasonic vibrator for ultrasonic treatment, so that the mixture is evenly dispersed in the mold. Finally, place the mold at room temperature for 24 h to allow the mixture to fully react with the moisture in the air to achieve the purpose of foaming. Place the foamed product in a blast drying box and dry it at 35 °C for 2 h. Then repeat the above steps to cast another layer on the foam layer. Finally, after the three layers of foam are poured and foamed, transfer them to in a blast drying box for drying at 35 °C for 5 h.

### 2.4. Characterization

The morphology of the CNTs@GN@Fe_3_O_4_ nanoparticles and cross-sectional views of CNTs@GN@Fe_3_O_4_/PU composites were examined by an S-4800 field emission scanning electron microscope (FE-SEM, Tokyo, Japan) equipped with an energy-dispersive X-ray spectrometer (EDS, London, UK). Before testing, the CNTs@GN@Fe_3_O_4_/PU samples were freezed in liquid nitrogen for 5 min to achieve brittle fractured surfaces, which were further sputtercoated with gold for better microscale observation.

The crystal structure of samples were analyzed using the technique of X-ray diffractomer (XRD, Rigaku Ultima IV, Innsbruck, Germany) equipped with a Cu Kα irradiation (λ = 0.11542 nm) from 10° to 80° at a scan rate of 6° min^−1^. The magnetic properties of CNTs@GN@Fe_3_O_4_ nanoparticles with different CNTs contents were tested by using a 7404 vibrating sample magnetometer (VSM, LakeShore Crytronics Inc., Westerville, OH, USA) at a field of up to 15 kOe. The EC of composite foams were acquired according to the formula: *σ* = *L*/(*S* × *R*), where *L* and *S* represent the sample length and cross-sectional area, respectively.

### 2.5. Electromagnetic Property Measurement

The relative permittivity *ε_r_* (*ε_r_* = *ε′* − *jε*′′) and permeability *μ_r_* (*μ_r_ = μ′* − *jμ′′*) were measured, covering a frequency range from 2 to 18 GHz based on the transmission/reflection coaxial line method. The samples for testing were sliced into a toroidal shape with 3.04 mm inner diameter and 7 mm outer diameter at 1.5~2.5 mm thickness. To reveal the EMWA performance, the RL values as a function of complex permittivity (*ε_r_*), complex permeability (*μ_r_*), frequency (*f*), and absorber thickness (*d*), were calculated as follows [20,21]:

The multi-layer structure design of composite absorbing materials can be carried out using transmission line theory. The multilayer composite absorbing structure and its equivalent transmission line model are shown in Reference [22].

In Figure 3, the relative complex permittivity of each layer of material is *ε_rk_*, the relative complex permeability is *μ_rk_*, *Z*_0_ is the wave impedance of free space, *d_k_* is the thickness of the *k*-th layer. Each layer of material is equivalent to the characteristic impedance after the transmission line *Z_k_* is [23]:(1)$$Z_{k}=Z_{0}\sqrt{\frac{\mu_{rk}}{\varepsilon_{rk}}}$$

The input impedance of the *k*-th layer of the multilayer composite structure can be expressed as:(2)$$Z_{ink}=Z_{k}\frac{Z_{ink-1}+Z_{k}\tan\left(j\gamma_{k}d_{k}\right)}{Z_{k}+Z_{ink-1}\tan\left(j\gamma_{k}d_{k}\right)}$$

The propagation constant *γ_k_* is expressed as:(3)$$\gamma_{k}=j\frac{2\pi f}{c}\sqrt{\mu_{rk}\varepsilon_{rk}}$$

Therefore, the reflection loss (RL) value of the electromagnetic wave absorber composed of *k*-th layers is expressed as:(4)$$RL=20\log\left|\frac{Z_{ink}-1}{Z_{ink}+1}\right|$$

## 3. Results and Discussion

### 3.1. Analysis of the Microscopic Morphology and Electromagnetic Properties of Fe_3_O_4_/CNTs@Fe_3_O_4_/CNTs@GN@Fe_3_O_4_

The prepared Fe_3_O_4_/CNTs@Fe_3_O_4_/CNTs@GN@Fe_3_O_4_ composite particles were characterized by scanning electron microscope (SEM) and the test results are shown in Figure 4a–c. It can be inferred from the SEM image that the Fe_3_O_4_ particles are regular round, relatively uniform, and well-dispersed, without obvious agglomeration. Through randomly photographed broken Fe_3_O_4_ particles, it can be seen that the Fe_3_O_4_ particles prepared in this article are solid spheres. As shown in Figure 4b, the composite particles of CNTs@Fe_3_O_4_ resemble tadpoles. Among them, the diameter of Fe_3_O_4_ particles is about 220 nanometers, and the surface is not smooth, showing lines similar to the shell of litchi. Carbon nanotubes have a diameter of 9 nanometers and a length of about 1.5 um, indicating that the composite nanoparticles of CNTs@Fe_3_O_4_ have been successfully prepared, and it is expected that the nano-scale particles of this composite structure will exceed the Snoek limit [24]. From Figure 4c, it can be seen that the microscopic morphology of CNTs@GN@ Fe_3_O_4_ is generally sandwich-like. Fe_3_O_4_ particles grow uniformly on the upper and lower sides of the graphene nanosheets, and the carbon nanotubes are combined with Fe_3_O_4_ particles, similar to the tadpole-like mentioned above, indicating that the composite particles of CNTs @GN@Fe_3_O_4_ have been successfully prepared. According to the test result of the vibrating sample magnetometer (VSM), as shown in Figure 4d, the hysteresis loop of CNTs@Fe_3_O_4_ hybrid material is smaller than that of Fe_3_O_4_ nanoparticles, and the saturation magnetization value decreases with the increase of CNTs content [25], because CNTs are a non-magnetic material. It can be seen from the hysteresis regression line that the saturation magnetization (Ms) value of Fe_3_O_4_ reaches 74.73 emu/g, the saturation magnetization (Ms) value of 50CNTs@Fe_3_O_4_ reaches 65.8 emu/g, and that there is a no obvious remanence (Mr) or correction Coercivity (Hc), which verify that it has superparamagnetism, shows that hybrid materials also have excellent soft magnetic properties. In addition, the hysteresis loops of 70CNTs@Fe_3_O_4_ and 50C1G1, 50C2G1, and 70C1G1 almost overlap. Figure 4e is the XRD pattern of CNTs, Fe_3_O_4_, CNTs@Fe_3_O_4_, and CNTs @GN@Fe_3_O_4_ at room temperature and neutral pH. Compared with the XRD pattern of pure carbon nanotubes, it can be concluded that the sharp peaks of CNTs@Fe_3_O_4_ at 2θ = 30.1° and 43.1° are the diffraction peaks of carbon, which correspond to the carbon (002) and (100) crystals of carbon nanotubes, respectively. This proved the existence of carbon nanotubes in the product and its good crystallinity. Combined with the XRD standard card (JCPDS NO. 19-0629), the synthesized CNTs@Fe_3_O_4_/CNTs @GN@ Fe_3_O_4_ samples are at 30.1°, 35.4°, 43.1°, 53.4°, and 56.9° except for the peak at 2θ = 17°. All diffraction peaks at and 62.5° are in agreement with the (220), (311), (400), (422), and (511) crystal planes of Fe_3_O_4_. According to relevant reference, we can be sure the XRD peaks of pure materials are correct. The amounts of GN and CNT compared to Fe_3_O_4_ is too small. That is the reason why peaks of CNT and GN detected for pure materials are not detected for composites. In summary, it is shown that the sample is indeed a composite of ferroferric oxide, carbon nanotubes, and graphene [26,27].

### 3.2. Analysis of the Microscopic Morphology and Structure of Composite Materials

Figure 5 shows the cross-sectional SEM images of the samples at different magnifications, and the distribution interface of the cells can be clearly seen. As shown in Figure 5a, the size of the cells is relatively uniform, as they are 8–12 μm. The existence of this cell structure can enhance the multiple reflections of incident electric waves inside the sample, which is extremely beneficial to the energy dissipation of electromagnetic waves. It can be inferred from Figure 5b,c that the CNTs@Fe_3_O_4_ and CNTs@GN@Fe_3_O_4_ particles are relatively evenly distributed inside the matrix without obvious agglomeration. The size of the CNTs and GN in the matrix is different from the size in the composite particles because the CNTs and GN in the matrix are obviously embedded in the interior of the matrix, which indicates that the combination of the particles and the matrix is extremely great, and there is good interface compatibility. In Figure 5d,e, it can be observed that there is an obvious bonding layer at the junction of adjacent layers, with a thickness of about 50–70 μm. The bonding layer does not appear to be cracked, and there are no gaps between adjacent layers. This perfect interfacial adhesion is attributed to the excellent film-forming and adhesive properties of PU, as well as the autonomous penetration and diffusion of PU macromolecular chains during LBL casting. It shows that the layer-to-layer combination is very good by the method of casting. It can be clearly seen from Figure 5f that there are almost no pores or only tiny pores at the bonding zone. This may be due to the denser wave-absorbing filler at the bonding surface and the pressure between the layers in the LBL process, which makes it difficult for the PU at the bonding surface to foam, and finally produces a dense separation belt. It can be seen from the SEM image that the thickness of each layer is highly close, indicating that the thickness design of the three-layer structure is well-controlled. Figure 5g–i show that only C, Fe, and O elements are present in the sample.

### 3.3. Wave Absorbing Performance of Three-Layer Foam

Figure 6 shows the simulated RL curves of three-layer magnetic foams (S1, S2, S3, S4, S5, S6) with six different ratio structures in the frequency range 2–18 GHz and the corresponding 2DRL simulation diagrams. From Figure 6a, it can be seen that the sample of S1 has a maximum peak at a thickness of 3.0 mm, the RL_min_ value can reach −45.75 dB, and the effective bandwidth is 3.3 GHz (11.4–14.7 GHz). It can be seen from Figure 6c that the RL_min_ peak of the S2 sample appears at 3.0, the minimum reflection loss value can reach −52.67 dB, and the effective bandwidth is 5.5 GHz (8.2–13.7 GHz). It can be seen from Figure 6e that when the S3 sample is 3.0 mm thick, the RL_min_ value can reach −66.05 dB, and the effective bandwidth is 5.5 GHz (8.1–13.7 GHz). It can be seen from Figure 6g that the RL_min_ value of sample S4 can reach −64.9 dB when the thickness of sample S4 is 3.0 mm, and the effective bandwidth is 5.0 GHz (8.1–13.1 GHz). It can be seen from Figure 6i that when the sample S5 is 3.0 mm thick, the RL_min_ value can reach −65.38 dB, and the effective bandwidth is 5.3 GHz (6.9–12.2 GHz). It can be seen from Figure 6k that the minimum reflection loss value of the sample S6 is significantly reduced when the thickness is 3.0 mm, the RL_min_ value is only −22.67 dB, and the effective bandwidth is 3.9 GHz (7.0–11.0 GHz). When the thickness is increased to 4.0 mm, the best RL_min_ value is −61.03 dB, and the effective bandwidth is 2.9 GHz (5.1–8.0GHz). From Figure 6b,d,f,h, it can be seen that the thickness of the samples with S1, S2, S3, and S4 reflection loss RL <−10 dB are all 2.0–5.0 mm. In Figure 6i,l, it can be observed that the thickness of the sample with S5 and S6 RL <−10 dB is 1.5–5.0 mm, and the range is slightly larger than the previous four. It can be clearly seen that when comparing the six samples, the better comprehensive absorbing performance appears at a thickness of 3.0 mm.

### 3.4. Analysis of Electromagnetic Parameters and Absorbing Mechanism of Three-Layer Magnetic Foam

According to Maxwell’s equation, the dielectric constant (ε) consists of a real part (ε′) and an imaginary part (ε′′), and the permeability (μ) consists of a real part (μ′) and an imaginary part (μ′′) composition. The real part (ε′) and (μ′) represent the storage of electrical and magnetic energy, and the imaginary part (ε′′) and (μ′′) represent the dissipation capacity of electrical and magnetic energy. It can be seen from Figure 7a that the ε′ of the three-layer foam sample decreases with the increase in frequency, however with the increase of the ratio of CNTs and GN, ε′ increases from 9.74 to 11.58, which indicate that CNTs and GN can increase the dielectric loss of materials [28]. It can be observed from Figure 8b that the imaginary parts of the dielectric constants of the six samples all show fluctuation peaks in the range of 2–4 GHz, 8–12 GHz, and 14–18 GHz. When the frequency increases from 2 GHz to 18 GHz, all the values of ε′ and ε′′ show a downward trend, which is called the dispersion behavior [29,30,31,32]. According to Debye theory, these fluctuation peaks are mainly related to the polarization behavior of the material. Since ion and electronic polarization usually occurs in the ultra-high frequency range (103–106 GHz), this factor can be eliminated.

These peaks are mainly attributed to the interface polarization loss. Because there is air between layers and in the microporous structure, the conductivity of the absorbing filler is different from that of air and PU, which will produce the Maxwell-wager effect. A large amount of positive and negative charges gather at the interface of Fe_3_O_4_, CNTs@Fe_3_O_4_, CNTs@GN@Fe_3_O_4_ and the air, and the interface of PU, forming a dipole moment, causing polarization loss [33,34]. According to Figure 7c, the real part of the magnetic permeability of the three-layer sample is between 0.96 and 1.16, which drops sharply in the range 2–4 GHz, and then tends to level off. The imaginary part of the permeability also decreases with the increase of frequency. A strong resonance peak appears in the range 4–8 GHz, and there is also an obvious resonance peak in the range 10–14 GHz, which is caused by magnetic loss [13,35], this coupling effect increases with the addition of CNTs and GN. According to Aharoni’s theory [36], when the body magnet is in a non-uniform magnetic system, the movement of the magnetic moment system is non-uniform, which will cause multiple resonant absorption peaks in the imaginary part of the permeability [10]. The resonant peaks below 10 GHz are mainly derived from natural resonance [37,38], and the resonant peaks in the range 10–14 GHz are mainly derived from exchange resonance. This exchange resonance is mainly derived from the inhomogeneity of the micro- and nano-particle structure in the multilayer structure. It can be observed that the higher the proportion of CNTs or GN in the material, the more obvious this phenomenon is.

In order to further explore the loss mechanism of the three-layer structure foam material, we can observe the change trend of the tg δε and tg δμ values of the six samples through Figure 7e,f. The value of tg δε shows an upward trend with the change of frequency. The value of tg δμ has an obvious upward trend at the beginning, and then tends to decrease. In the range of 2–8 GHz, tg δμ is much larger than tg δε, indicating that in this frequency range, magnetic loss is the main mechanism. In the range of 8–18 GHz, tg δε is greater than tg δμ, indicating that the loss mechanism has changed from magnetic loss to dielectric loss.

It can be seen from Figure 8a that the absorption peaks of the six samples at 3.0 mm show that with the increase in the ratio of CNTs and GN in the second and third layers, the peaks gradually move to the low frequency region, S2, S3, S4, and S5. The effective absorption bandwidth of the four samples can cover almost the entire X-band (8–12 GHz), which is of great significance to the design of radar absorbing materials. One can observe the impedance matching of each sample from the normalized impedance *Z_in_*/*Z*_0_, when the impedance of the material (*Z_in_*) is close to the free impedance of space (*Z*_0_). That is, the closer the value of *Z_in_*/*Z*_0_ is to 1, the better its impedance matching degree, which indicates that electromagnetic waves enter the material more easily and the electromagnetic wave absorption ability is better. The impedance ratio of S1 in the high frequency range is closer to 1, and S2, S3, S4, and S5 are closer to 1 in the low frequency range. S6 has the worst absorption effect, so its impedance matching value is far away from 1 in a long frequency range. It can be concluded that with the increase of the ratio of CNTs and GN in the second and third layers, the impedance ratio of each sample close to 1 gradually shifts to the low frequency region, which is consistent with the movement of the wave crest. It shows that through the design of multi-layer structure, the absorbing effect can be effectively improved, and the peak value can be shifted to the low frequency region.

Electrical conductivity is a vital manifestation of electromagnetic wave absorption performance. The electrical conductivity of the composite material is calculated according to formula $\sigma =\frac{L}{S}+\frac{1}{R}$ (where *L* is the length of the test sample, *S* is the cross-sectional area of the sample, and *R* is the resistance of the sample). From the point of view of electrical conductivity, the electrical conductivity values of S1, S2, S3, S4, S5, and S6 are 4.89 S/cm, 6.58 S/cm, 9.39 S/cm, 8.21 S/cm, 9.57 S/cm, and 10.33 S/cm. From the trend of the RL value of the material mentioned above, it can be seen that high conductivity will not always enhance the electromagnetic wave absorption performance of the material. Because the high conductivity will reduce the impedance matching, it will prevent the microwave from entering the inside of the material, which is why the conductivity of S6 is the highest, but the microwave absorption performance is actually reduced.

Considering the great heterogeneity of the sample, some statistical tools were used to evaluate the significance of data. It is hard to list all dates, as the most representative data are the minimum of the RL value, the effective bandwidth, and the abscissa of the point when it intersects the line RL = −10. The above data came from five experiments when the thickness was 3 mm, as shown in Figure 9a. Using the statistical tool to evaluate the significance of the date, as shown in Figure 9b. The *p*-value is 0.747 > 0.05 and F = 0.484 < F crit rate, and it indicates that there is no marked difference between the five groups.

Therefore, three conclusions can be drawn: Firstly, to a certain extent, the increase in electrical conductivity helps to improve the electromagnetic wave absorbing ability of composite materials, because the introduction of CNTs and GN can produce numerous conductive paths inside the foam matrix [17]. According to the reference, GN and CNTs have fabulous conductivity and thermal conductivity. The resistance loss mechanism provided by their dipole polarization is conducive to the transfer of electrons [26]. A certain amount of CNTs will increase the degree of cross-linking. GN has a larger specific surface area. After compounding with Fe_3_O_4_ and CNT, it will form multiple interfaces, resulting in multiple reflections of electromagnetic waves, which is conducive to the absorption of electromagnetic waves. The micro-current is enhanced between the conductive networks. Therefore, after electromagnetic waves enter the interior of the composite material, the highly conductive composite material will more easily form an internal current inside, thereby converting more electromagnetic wave energy into internal energy for consumption, and increasing the electromagnetic wave absorption loss of the material [39]. Secondly, the conductivity is too large, which will make the material tend to be shielded material, thereby reducing the impedance matching, resulting in a decrease in the microwave absorption effect; Finally the analysis of variance shows that the stability of the experiment is great. There is no marked difference between the five groups.

The mechanism of the three-layer electromagnetic foam effectively increasing the absorption strength and effective bandwidth lies in the following aspects, as shown in Figure 10. Firstly, the introduction of the impedance matching layer reduces the reflection of electromagnetic waves on the surface of the material, allowing more incident electromagnetic waves to enter the interior of the substrate. Secondly, Fe_3_O_4_, CNTs@Fe_3_O_4_, CNTs@GN@Fe_3_O_4_ three kinds of wave absorbers bring different loss effects, such as the magnetic loss of Fe_3_O_4_, the dielectric loss of CNTs and GN, between particles and particles, and between particles and matrix. The polarization of the interface will increase the loss of electromagnetic waves. Finally, and the most significant point, the design of the multi-layer structure increases the reflection path of the electromagnetic wave inside the substrate, forcing the electromagnetic wave to reflect and penetrate multiple times in the material, and then further attenuate the electromagnetic wave.

## 4. Conclusions

Fe_3_O_4_/CNTs@Fe_3_O_4_/CNTs@GN@Fe_3_O_4_ magnetic nanoparticles were prepared by a simple solvothermal method and dispersed in one-component moisture self-foaming polyurethane. A series of three-layer magnetic foams were prepared by the LBL layered pouring method. Through the analysis of the foam morphology, electromagnetic parameters, and absorbing properties, it is found that the multilayer structure has an important influence on the electromagnetic absorbing properties of the magnetic foam. The research results justify that the electromagnetic wave absorption performance of the three-layer foam at a thickness of 3.0 mm is more excellent. The main manifestation is that the RL_min_ values of S2, S3, S4, and S5 are between 52.0 and 67.0 dB, and the effective bandwidth is above 5.0 GHz and can cover the entire X-band. All this is due to the synergy of dielectric and magnetic loss between Fe_3_O_4_, CNTs and GN, which improves the performance of EMWA; The multilayer and microporous structure can increase the interface polarization and the path of electromagnetic wave reflection and scattering, and further increase the consumption of incident EM. The results show that electrical conductivity is also a major manifestation of the electromagnetic wave absorption performance of magnetic foam. To a certain extent, the increase in electrical conductivity helps to improve the electromagnetic wave absorbing ability of the composite material, which is mainly manifested in the enhancement of the 3D conductive network inside the foam. This contributes to electron transfer, enhances the micro-current, and converts EM into internal energy for consumption.

## Figures and Tables

**Figure 1 materials-14-07244-f001:**
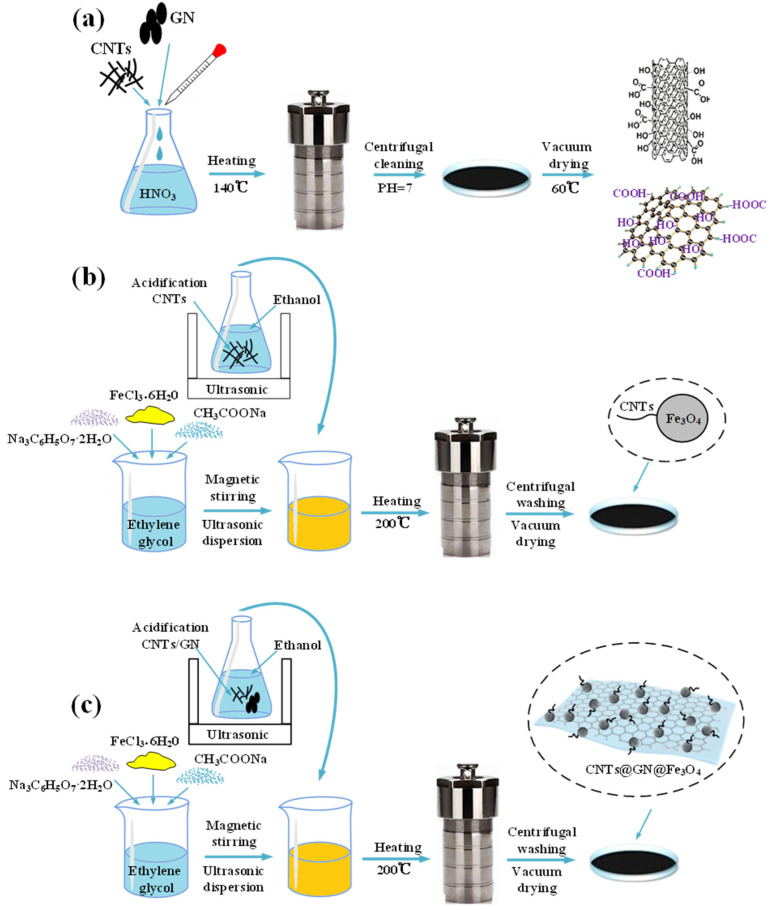
(**a**) Acidification process of CNTs and GN; (**b**) preparation process of CNTs@GN@Fe_3_O_4_ particles; (**c**) preparation process of CNTs@GN@Fe_3_O_4_/PU composite material.

**Figure 2 materials-14-07244-f002:**
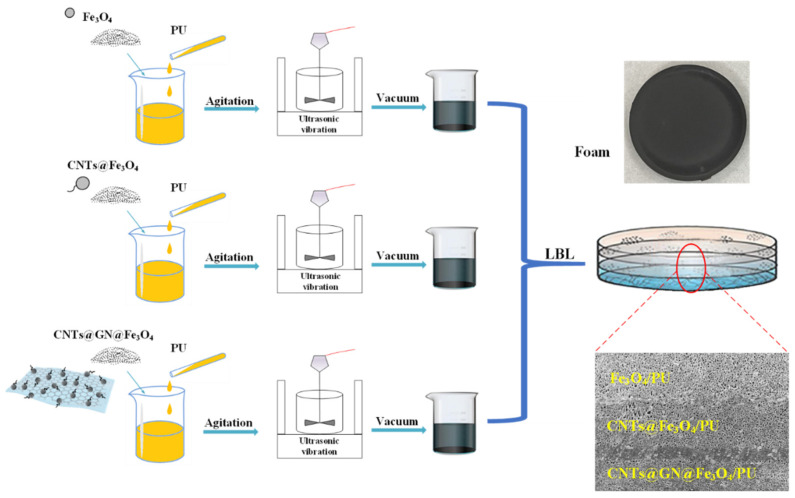
Preparation of three-layer composite.

**Figure 3 materials-14-07244-f003:**
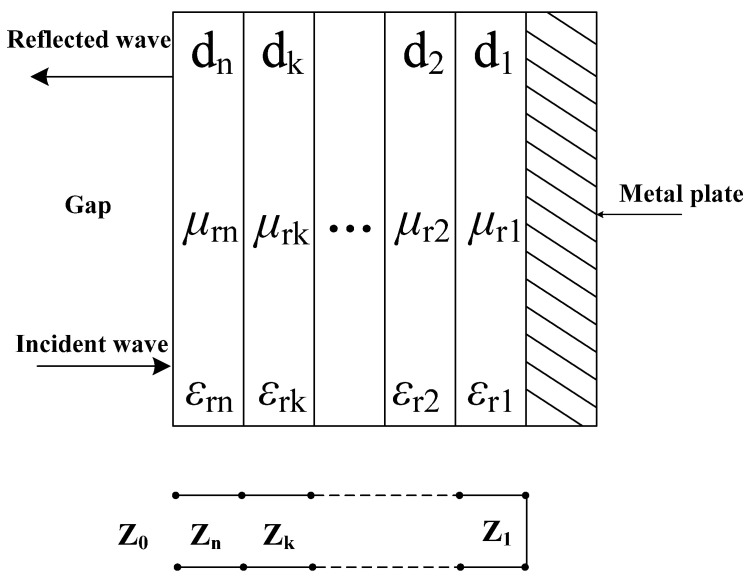
Schematic diagram of the absorbing structure composed of multilayer materials; equivalent transmission line model of the multilayer composite structure.

**Figure 4 materials-14-07244-f004:**
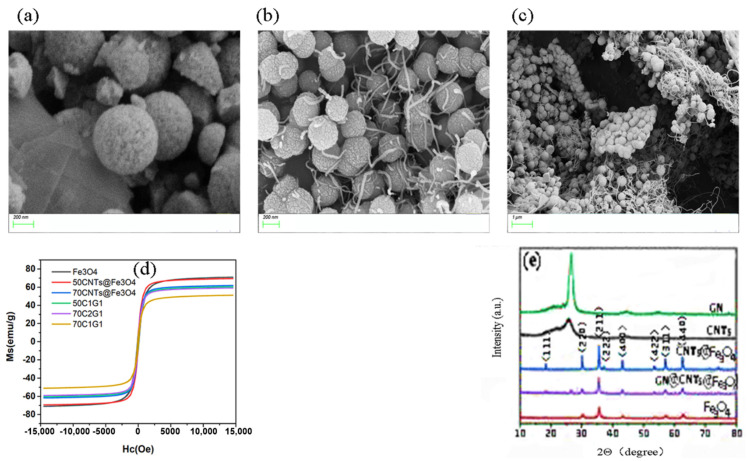
(**a**–**e**)The characterization results of the Fe_3_O_4_/CNTs@Fe_3_O_4_/CNTs@GN@Fe_3_O_4._

**Figure 5 materials-14-07244-f005:**
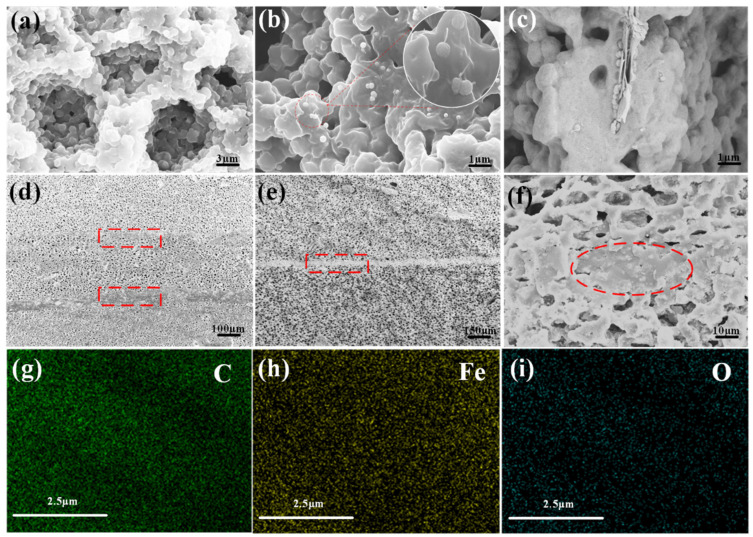
SEM images of cross-sections of samples under different magnifications (**a**) SEM image of Fe_3_O_4_/PU at 5000× magnification; (**b**) SEM image of CNTs@Fe_3_O_4_/PU at 8000× magnification; (**c**) SEM image of CNTs@GN@Fe_3_O_4_/PU at 6000× magnification; (**d**) SEM image of three-layer foam at 50× magnification; (**e**) SEM image of three-layer foam at 100× magnification; (**f**) SEM image of three-layer foam at 1000× magnification; (**g,h,i**) element mapping.

**Figure 6 materials-14-07244-f006:**
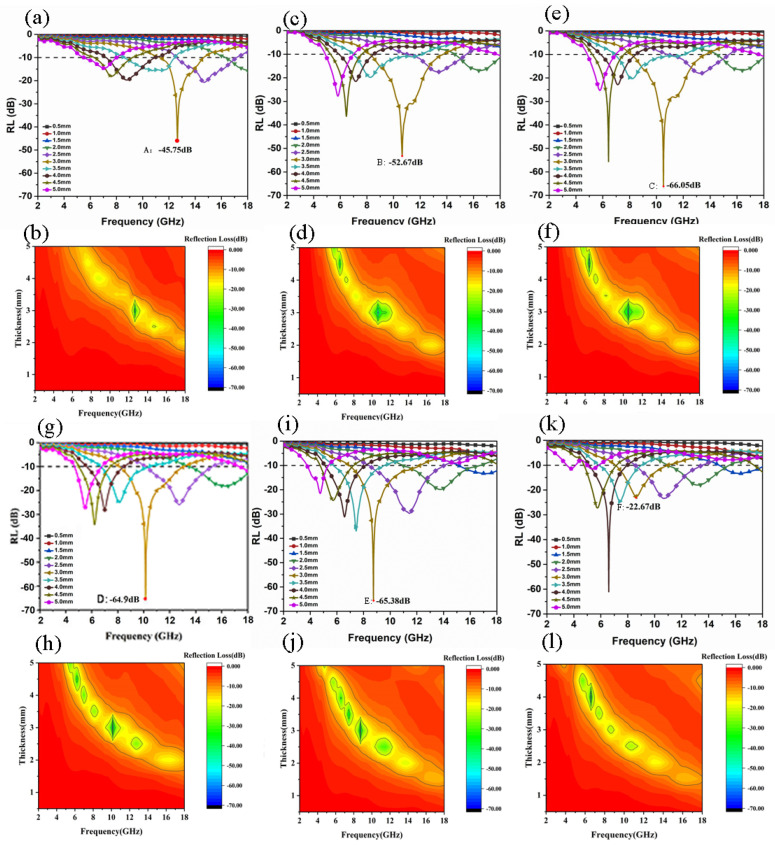
The simulated RL curves of three-layer magnetic foams (**a**,**b**) S1, (**c**,**d**) S2, (**e**,**f**) S3, (**g**,**h**) S4, (**i**,**j**) S5, and (**k**,**l**) S6 electromagnetic absorption RL curve diagram and corresponding 2D simulation diagram.

**Figure 7 materials-14-07244-f007:**
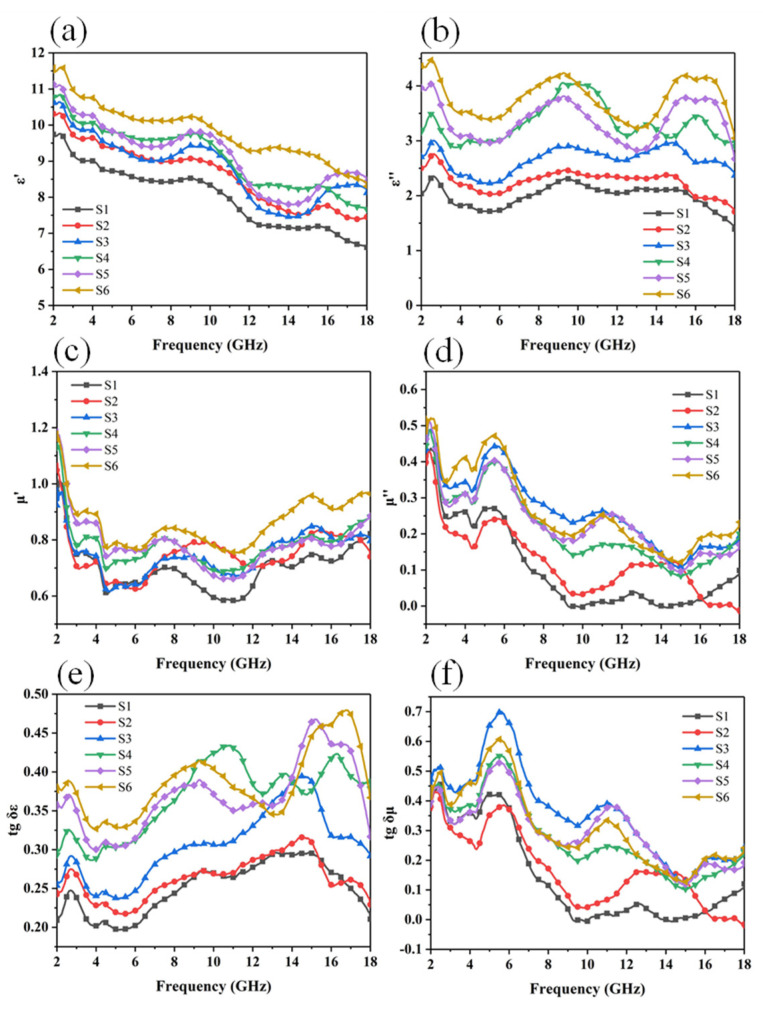
(**a**) ε′ change curve within 2–18 GHz of three-layer magnetic foam; (**b**) ε′′ change curve; (**c**) μ’ change curve; (**d**) μ′′ change curve; (**e**) tgδε change curve; (**f**) tg δμ change curve.

**Figure 8 materials-14-07244-f008:**
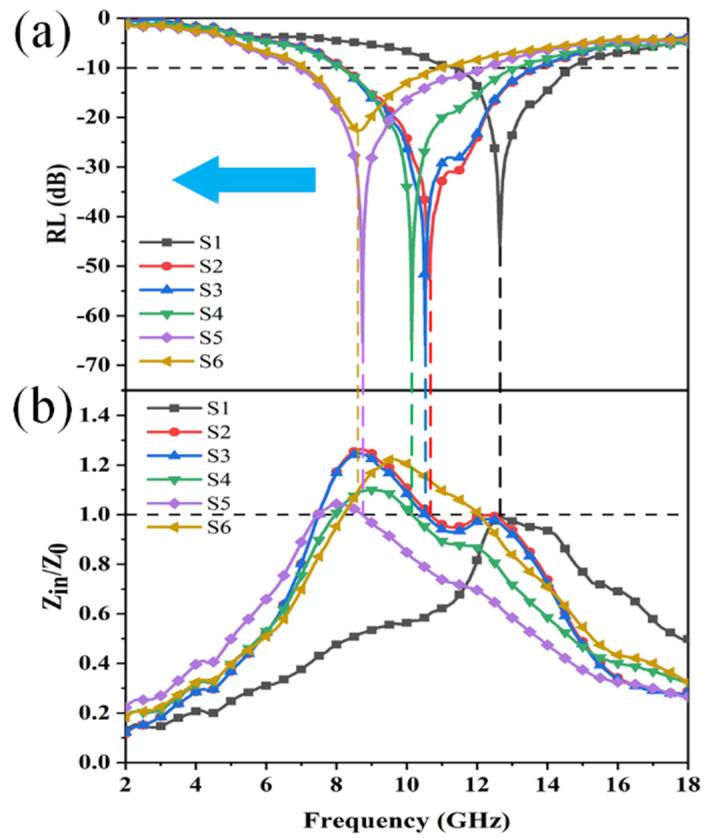
(**a**) RL value of six kinds of three-layer magnetic foam at a thickness of 3.0 mm; (**b**) *Z_in_*/*Z*_0_ value of six kinds of three-layer magnetic foam at a thickness of 3.0 mm.

**Figure 9 materials-14-07244-f009:**
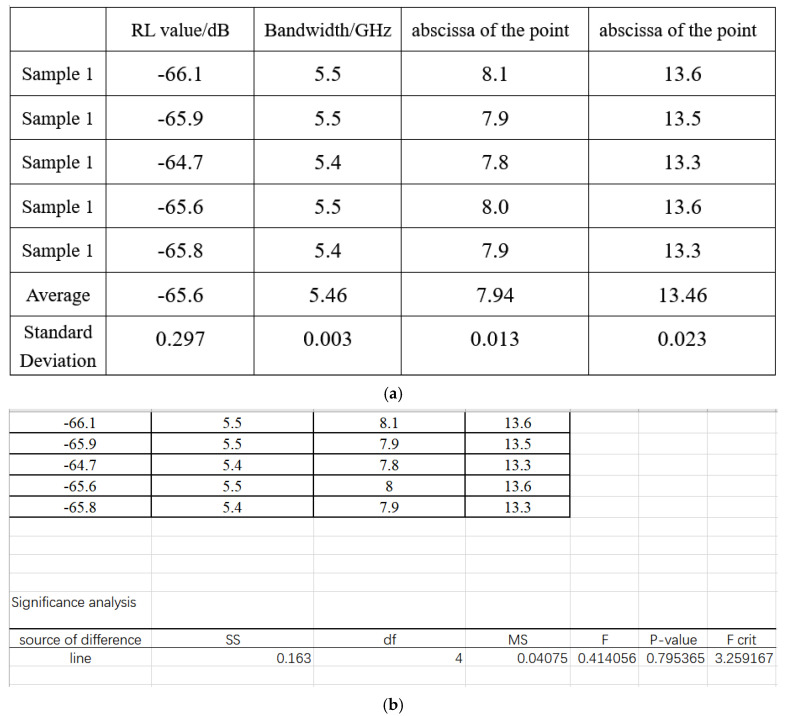
(**a**) the average value and the standard deviation of five samples. (**b**) Significance analysis of five samples.

**Figure 10 materials-14-07244-f010:**
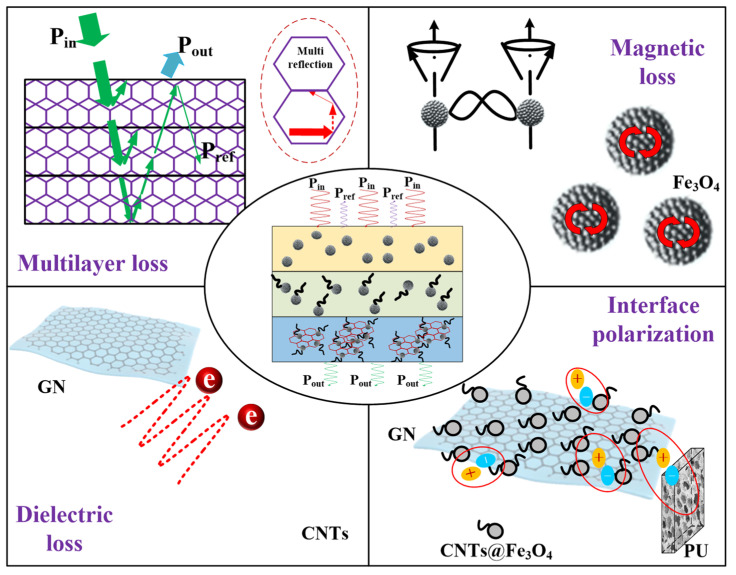
Electromagnetic wave absorption mechanism of three-layer electromagnetic foam.

## Data Availability

Not applicable.

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
