# Peer review of "Electromagnetic Wave Absorption and Mechanical Properties of CNTs@GN@Fe3O4/PU Multilayer Composite Foam"

_materials, 2021, doi:10.3390/ma14237244_

Round 1
Reviewer 1 Report
This is a good article that might be suitable for publication after taking into account the following suggestion, which will certainly improve this manuscript.
- Abstract is too long and contains mostly literature review. Please formulate more clearly all new results that were not published before even by other groups.
- Line 38. This sentence needs very general (review) references, while [1]-[3] – are not !!! See as an example: Deruelle, F. (2020). The different sources of electromagnetic fields: dangers are not limited to physical health. Electromagnetic biology and medicine, 39(2), 166-175.
- Line 40, 45, 54 need supporting references.
- Line 58-69. Please, give more motivational explanations why Fe3O4? Why Fe3O4 and not Fe2O3.
- For a wider range of readers, more information on preparation methods and applications would be desirable. References [9-10], which are of course important and interesting, are not the most recent. However, there are many new and promising uses. See for examples, some of them published this year in MDPI journals,
Serga, V.et al . Impact of Gadolinium on the Structure and Magnetic Properties of Nanocrystalline Powders of Iron Oxides Produced by the Extraction-Pyrolytic Method. Materials 2020, 13, 4147.
Li, Y.; Wang, Z.; Liu, R. Superparamagnetic α-Fe2O3/Fe3O4 Heterogeneous Nanoparticles with Enhanced Biocompatibility. Nanomaterials 2021, 11, 834.
- Line 182, Figure 3. It would be very helpful to provide typical thickness values.
- Line 307 “Because there is air between layers and in the microporous structure, the ..” Could you comment here on are there any effects associated with uncontrolled adsorption of gas molecules on the surface?
- Line 329. Figure 7. Could you comment on whether there is aging of the samples after such manipulations?
- Conclusion: Please, formulate new results and observations more clearly in terms of physics.
Author Response
Comment1:
1.Abstract is too long and contains mostly literature review. Please formulate more clearly all new results that were not published before even by other groups.
Response:
We have modified abstract and formulate more results which we found.
- Line 38. This sentence needs very general (review) references, while [1]-[3] – are not !!! See as an example: Deruelle, F. (2020). The different sources of electromagnetic fields: dangers are not limited to physical health. Electromagnetic biology and medicine, 39(2), 166-175.
Response:
We add the reference 4、5 and 6 which is more detailed.
- Line 40, 45, 54 need supporting references.
Response:
We add the relevant reference.
- Line 58-69. Please, give more motivational explanations why Fe3O4? Why Fe3O4 and not Fe2O3.
Response:
Due to the difference of their crystal structure,we decide to use Fe3O4.
Generally,iron has three oxides——α- Fe2O3、γ- Fe2O3 and Fe3O4. α- Fe2O3 is a stable corundum structure, oxygen ions are the most densely packed hexagonal, and iron ions are in the middle of the two cation layers. The arrangement of iron ions makes the distance between them the largest of the three oxides. Generally speaking, this arrangement has the strongest chemical stability.
In terms of crystal structure, Fe3O4 and γ- Fe2O3 belongs to spinel structure, but in the face centered cubic lattice composed of divalent oxygen ions, their conductivity is obviously different due to the different valence states of iron ions
Fe3O4 belongs to anti spinel configuration, and its structural formula can be expressed as So divalent iron ions and some trivalent iron ions occupy the gap of octahedron composed of oxygen ions, it is possible to exchange electrons in the lattice, so the resistivity is small. And γ- Fe2O3 belongs to orthospinel configuration, and the structural formula can be expressed as (□ represents the gap). Trivalent iron ions occupy the tetrahedral and octahedral gap positions formed by oxygen ions, and there are a certain number of vacancies at the octahedral position. There are no exchangeable electrons, so the resistivity is relatively large. In the microwave incident band, Fe3O4 can produce both magnetic loss and electrical loss to electromagnetic waves. It is a double complex dielectric material.That is the reason why Fe3O4.
- For a wider range of readers, more information on preparation methods and applications would be desirable. References [9-10], which are of course important and interesting, are not the most recent. However, there are many new and promising uses. See for examples, some of them published this year in MDPI journals,
Response:
We decide to add an example which in MDPI journals.
‘Song used simple hydrothermal method to prepared MWCNT / Fe3O4 composites . Ferromagnetic matrix composites with light weight, controllable morphology and strong microwave absorption ability were prepared by adjusting particle size and holding time. Through the selection of functional groups and solutions, the directional agglomeration of Fe3O4 is realized to form a complex network structure. Finally, when the solvothermal time is 15 h and the size of microspheres is 400 nm, the reflection loss is - 38 dB’
- Line 182, Figure 3. It would be very helpful to provide typical thickness values.
Response:
We add Figure 4 to show some typical thickness values.
Figure 4
- Line 307 “Because there is air between layers and in the microporous structure, the ..” Could you comment here on are there any effects associated with uncontrolled adsorption of gas molecules on the surface?
Response:
The adsorption of uncontrolled gas molecules on the surface will produce a layer of medium with dielectric constant different from air and PU,which will produce the Maxwell-wager effect.
- Line 329. Figure 7. Could you comment on whether there is aging of the samples after such manipulations?
Response:
Will not.
- Conclusion: Please, formulate new results and observations more clearly in terms of physics.
Response:
We have changed some expression.

Reviewer 2 Report
The manuscript presents results of investigations related to very important technical area. While this 'technical' importance is quite clear the academic value of the work is questionable. However described results may be interesting for readers and thus publication of the manuscript may be acceptable after some modifications.
First of all the authors must reread own manuscript to eliminate evident mistakes and uncertainties:
- there are no results on mechanical properties investigation and thus mentioning of these properties must be removed from title and from Section 2.4;
- also there are no results on Raman spectrsocopy and thus mentioning of RS measurements must be removed from Section 2.4;
- there are two different abbreviations for electromagnetic wave absorbing materials (EMWA and EMW) but no definitions for EWAP, FCC, FLC. Please add.
- check and modify expressions like: '...pollution has become one of the pollution...' and '...shielding and wave absorption; And combined with one/two-dimensional carbon..' in Abstract, '...the size of the cells is relatively uniform, with a size of about 8-12um..' at page 8, etc
- replace 'certain amount' and 'certain mass' to exact values;
- please try to modify sample names because in present form they look very cumbersome that creates problems for text readig;
- 'Figure 4d' at line 222 on page 7 should be replaced probably on Figure 4e?
- to which sample belongs XRD spectrum marked as rGO@CNTs@Fe3O4 in Figure 4? There is no such samples in the nomenclature.
Also, please, indicate suitable references justifing asseptance of XRD peaks to CNT, GN, Fe3O4 (page 7). And explain why XRD peaks of CNT and GN detected for pure materials are not detected for composites.
Please explain what does it means (page 13): 'As we all know, electrons are transported in the CNTs and GN layers through electron migration or jumping.' I do not know about this elelctron transport.
It is also necessary clarify following stetement: ' GN has a larger specific surface area, which provides opportunities for electrons to jump through different interfaces.' How surface area effect on electron jumping?
Author Response
Comment2:
- there are no results on mechanical properties investigation and thus mentioning of these properties must be removed from title and from Section 2.4;
Response:
We have deleted it.
also there are no results on Raman spectrsocopy and thus mentioning of RS measurements must be removed from Section 2.4;
Response:
We have deleted it.
there are two different abbreviations for electromagnetic wave absorbing materials (EMWA and EMW) but no definitions for EWAP, FCC, FLC. Please add.
Response:
We have added it.
check and modify expressions like: '...pollution has become one of the pollution...' and '...shielding and wave absorption; And combined with one/two-dimensional carbon..' in Abstract, '...the size of the cells is relatively uniform, with a size of about 8-12um..' at page 8, etc
Response:
We have improved the English of the article.
- replace 'certain amount' and 'certain mass' to exact values;
Response:
We have replace the word to exact values in article.
please try to modify sample names because in present form they look very cumbersome that creates problems for text reading;
Response:
We refer to other literature,decide not change the expression.
'Figure 4d' at line 222 on page 7 should be replaced probably on Figure 4e?
Response:
Yes. We have changed it.
to which sample belongs XRD spectrum marked as rGO@CNTs@Fe3O4 in Figure 4? There is no such samples in the nomenclature.
Response:
That is an incorrect name of sample marked as rGO@CNTs@Fe3O4 in figure 4.We have replace the name in the figure 4.
Also, please, indicate suitable references justifing asseptance of XRD peaks to CNT, GN, Fe3O4 (page 7). And explain why XRD peaks of CNT and GN detected for pure materials are not detected for composites.
Response:
We have added some reference to justifing asseptance of XRD peaks to CNT, GN, Fe3O4.According to the reference,we think the amounts of GN and CNT comparing with Fe3O4 is too small. That is the reason why peaks of CNT and GN detected for pure materials are not detected for composites.
- Please explain what does it means (page 13): 'As we all know, electrons are transported in the CNTs and GN layers through electron migration or jumping.' I do not know about this elelctron transport.
Response:
We decide to change the sentence to ‘According to the reference GN and CNTs have good conductivity and thermal conductivity. The resistance loss mechanism provided by their dipole polarization is conducive to the transfer of electrons.’
It is also necessary clarify following stetement: ' GN has a larger specific surface area, which provides opportunities for electrons to jump through different interfaces.' How surface area effect on electron jumping?
Response:
We decide to change the sentence to ‘GN has a larger specific surface area. After compounding with Fe3O4 and CNT, it will form multiple interfaces, resulting in multiple reflections of electromagnetic waves, which is conducive to the absorption of electromagnetic waves.’
Reviewer 3 Report
Authors described a simply study on homo-inhomogenous deformation of alloy specimen. Even if this paper is not so innovative, the topic is quite interesting but still too
As general consideration, the language is very poor. It is very hard to read this manuscript. The style is far to reach the level of acceptability.
Please check the uncertainties in the secondo column. Two digits are too much. Furthermore, marked the values significantly different from each other with a capital letter.
The main issue (excluding the style) is about the data treatment. Authors did not report any statistical data. Considering the great heterogeneousity of the materials tested, authors must replicated each experiment, reporting the average value and the uncertainty. Afterwards, the run statistically tools to evaluate the significance of their data.
Considering the points raised above, I cannot endorse the publication of this work in the present shape. I require at least major revisions.
Author Response
We have modified the figure、marked the value with a capital letter and check the uncertainties. We also have added figure 10a and 10b to report the average value, standard deviation, P-value and F crit to evaluate the significance of date. The result shows no marked difference between the date. Finally,we have embellished the article to make it easier to read and understand.
Round 2
Reviewer 2 Report
The revised manuscript is accptable for publication.
Reviewer 3 Report
it is now accetable
This manuscript is a resubmission of an earlier submission. The following is a list of the peer review reports and author responses from that submission.
Round 1
Reviewer 1 Report
The manuscript describes creation of a polyurethane based composite material containing carbon nanotubes, nanographite fragments and Fe2O3 particles. Material is created and investigated as electromagnetic wave absorber. Similar applications of different nano-carbon materials are reported in numerous publications and novelty of present work is not evident in view of this.
The most important problem which is not discussed in the manuscript with suitable details is mechanism of electromagnetic wave absorption - how nanometers scale structures effect on abosrption of 3 cm long waves? It is mentioned in the manuscript importance of 'defects' and 'polarization' but it remains not clear what kind of deffects are assumed here? The composite is quite desordered and, thus, completely 'defective'. Possible structural defects of nano-carbon composites are not discussed and their effect on absorption is not clear.
There are many technical drawbacks in the manuscript requiring major revision before further consideration of the manuscript acceptance for publication.
- Abbreviations are used before or without their explanations (e.g. PU, rlmin in Abstract)
- It is not explained what mean used names for samples (50C1G1, 50C1G2, 50C1G2, C1G1, 70C1G1, etc.) and what is difference between these samples.
- What kind of carbon nanotubes were used Single Wall or Multi-Walled?
- Usage of 'graphene' name for 10 nm thick particles is very questionably because they contain about 30 atomic layers and their properties quite different from true (single layer) graphene.
- Used 'the transmission/reflection coaxial line method' must be described in details.
- There are numerous typos and 'strange' expressions like 'the diffraction peaks of carbon', 'In the matrix, it shows that the combination of particles and matrix is very good, and there is good interface compatibility.', 'Due to the strong π-π bond interaction between GN, CNTs and PU, it is beneficial to the charge transfer between GN, CNTs and PU, and increases the dielectric constant value of the composite material, thus greatly increasing the electrical loss of the material.', 'Therefore, the absorption of electromagnetic waves by CNTs@GN@ Fe3O4/PU composite material is the effect of the synergistic effect of dielectric loss 336 and magnetic loss.' There is no meaning in some of these expression or contains incorrect statements.
- Presentation of the dielectric constant (ε) and the permeability (μ) as sum of a real and imaginary part is follows from their definition but not 'According to Maxwell’s equation...' (see Section 3.4). Instead of numerous references mentioned here the authors should use a suitable textbook.
- Fig. 7 presents and corresponding text presents very speculatively and not suitably assumed mechanism of electromagnetic waves absorption.
In general, mentioned 'technical' drawbacks make it difficult (and even impossible in some parts) understanding of the manuscript.
Author Response
- Abbreviations are used before or without their explanations (e.g. PU, rlmin in Abstract)
Response: polyurethane(PU), minimum reflection loss(RLmin)
- It is not explained what mean used names for samples (50C1G1, 50C1G2, 50C1G2, C1G1, 70C1G1, etc.) and what is difference between these samples.
Response: The total amount of CNTs and GN added is 50mg and 70mg, and the ratio of CNTs and GN is changed to make the mass ratio 1:1 (C1G1), 1:2 (C1G2) and 2:1 (C2G1) to obtain Six powder samples were named 50C1G1, 50C1G2, 50C2G1, 70C1G1, 70C1G2 and 70C2G1.
- What kind of carbon nanotubes were used Single Wall or Multi-Walled?
Response: Multi-walled carbon nanotubes
- Usage of 'graphene' name for 10 nm thick particles is very questionably because they contain about 30 atomic layers and their properties quite different from true (single layer) graphene.
Response: graphene nanosheets
- Used 'the transmission/reflection coaxial line method' must be described in details.
Response:
In the formula, and are the complex permeability and the complex permittivity of the material, respectively. The real part of the permeability and the real part of the dielectric constant represent the storage of the incident electromagnetic wave magnetic field energy and electric field energy Ability; and the imaginary part of permeability and the imaginary part of permittivity reflect the loss ability of electromagnetic waves.
Good impedance matching is the first principle to be followed when designing a wave-absorbing coating. The impedance matching coefficient Z is a parameter to measure the impedance matching of the wave-absorbing coating, which can be calculated by the following formula:
|
(2.7) |
Where: is the input impedance of the absorbing material; is the frequency of the incident electromagnetic wave; d is the thickness of the absorbing material; c is the propagation rate of electromagnetic waves in vacuum; h is the Planck constant; is the impedance of free space, and are the permeability and permittivity of free space respectively;
Generally, the index to evaluate the electromagnetic absorption performance of a material is the reflection loss (RL) value. When the value of RL is less than -10dB, it means that 90% of the incident microwave is absorbed by the material, which is up to the standard in practical applications. When the material thickness and the tested frequency range are known, the value of RL can be calculated by transmission line theory, the calculation formula is as follows:
At present, the electromagnetic wave absorption performance of most composite materials is measured by direct measurement method, which has the advantages of fastness, accuracy and wide application range. The direct measurement method uses a vector network analyzer (VNA) to measure the electromagnetic characteristics of the sample. The principle is to use a coaxial cable to transmit the plane electromagnetic wave emitted by the vector network analyzer signal source. When the plane electromagnetic wave passes through the sample to be tested, The analyzer calculates the reflected and transmitted electromagnetic wave power density to obtain the scattering parameter (S parameter).
According to the transmission/reflection model, the two-port network transmission coefficient T of the vector network analyzer can be expressed as:
The reflection coefficient is:
According to the Nicolson algorithm, the sum and difference of the S parameters are recorded as:
definition:
The function relationship between the transmission coefficient T and the S parameter is:
At this time, let:
Combining the above formulas, the expressions of complex permittivity and complex permeability can be obtained:
Then substituting the calculated complex permittivity () and complex permeability () of the sample to be tested, the electromagnetic absorbing efficiency of the sample to be tested can be calculated. At present, the electromagnetic performance test of materials generally adopts the coaxial method, the waveguide method and the arch method.
Figure Transmission/reflection model diagram
- There are numerous typos and 'strange' expressions like 'the diffraction peaks of carbon', 'In the matrix, it shows that the combination of particles and matrix is very good, and there is good interface compatibility.', 'Due to the strong π-π bond interaction between GN, CNTs and PU, it is beneficial to the charge transfer between GN, CNTs and PU, and increases the dielectric constant value of the composite material, thus greatly increasing the electrical loss of the material.', 'Therefore, the absorption of electromagnetic waves by CNTs@GN@ Fe3O4/PU composite material is the effect of the synergistic effect of dielectric loss 336 and magnetic loss.' There is no meaning in some of these expression or contains incorrect statements.
We have improved the description in the reviewed sample.
- Presentation of the dielectric constant (ε) and the permeability (μ) as sum of a real and imaginary part is follows from their definition but not 'According to Maxwell’s equation...' (see Section 3.4). Instead of numerous references mentioned here the authors should use a suitable textbook.
Response: Liang Canbin, Qin Guangrong, Liang Zhujian. General Physics Course. Electromagnetism. Beijing: Higher Education Press, 2012.12: 377, 378
- Fig. 7 presents and corresponding text presents very speculatively and not suitably assumed mechanism of electromagnetic waves absorption.
The figure shows the electromagnetic wave absorption mechanism of CNTs@GN@Fe3O4/PU magnetic foam. The excellent electromagnetic wave absorption ability of the composite material comes from the combined action of multiple absorption mechanisms. First of all, the microporous structure inside PU can increase the reflection interface of electromagnetic waves, resulting in multiple scattering and reflection. In addition, the introduction of GN can also increase this effect. Because of its layered structure, it can act as another reflection source. And because of its larger specific surface area, it makes the migration of electrons easier. Secondly, the high carrier mobility and conductivity characteristics of GN and CNTs are derived from the hybridization of sp2 carbon atoms. Under the action of an electromagnetic field, following the free electron theory, micro-currents will be generated, which will convert electromagnetic waves into internal energy dissipation. Third, the introduction of GN increases the defects, disorder, and the amount of functional groups within the material, which will make it easier to form dipoles. As the frequency of the electric field increases, the change of the dipole cannot keep up with the change of the electric field frequency. This hysteresis phenomenon [106] can cause polarization and related relaxation, which is conducive to the loss of electromagnetic waves. Fourth, the Fe3O4 particles are attached to the surface of the GN, so that the interface polarization is provided between GN/Fe3O4, CNTs/Fe3O4 and CNTs/GN, which enhances the interface relaxation due to the dielectric loss of the hybrid material. Form a structure similar to a capacitor. Finally, as a ferromagnetic material, Fe3O4 has magnetic loss, natural resonance and eddy current loss, and further loss of electromagnetic waves entering the material.
Reviewer 2 Report
The paper about the EM shielding herein considered is quite interesting evenif it is not so innovative.
There are some issues to solve.
Please do not use acronyms in the abstract.
At line 41, what did you mean with "limited EMWA performance adjustment ability et al, which hardly meet the increasing"?
At line 334 change "This is because" with "This is due to"
Data reported in figure 8 are poorly commented. You should extrapolate Young'modulus, UTS and maximum elongation from the stress vs strain curves.
Additionally, i have soume doubts about the methods adopted. Were teh measurments repeated? In any case you must provide data with the related uncertainty values.
Furthermore, i suggest to add a section where the performances of teh material herein reported is compared with literature data. This will add a significant information to the manuscript.
As it is, this paper is not acceptable and major revisions are required.
Author Response
1.Please do not use acronyms in the abstract.
As shown in the reviewed version.
- At line 41, what did you mean with "limited EMWA performance adjustment ability et al, which hardly meet the increasing"?
Response:The electromagnetic (EM) interference and radiation, originated from the swift evolution of information technology, have pernicious impacts on the electronic instrument and human health. For eliminating EM pollution, consistent efforts have been undertaken to exploit novel, highly efficient electromagnetic wave absorption (EMWA) materials which can safely dissipate the EM energy into heat or other energy without secondary EM wave radiation. Ordinary metal based EMWA materials often suffer from the weaknesses including poor flexibility, high density and limited regulation ability of EMWA performance, which barely satisfies the increasing demands from the next-generation flexible electronics
- At line 334 change "This is because" with "This is due to"
We have changed the description "This is because" with "This is due to".
- Data reported in figure 8 are poorly commented. You should extrapolate Young'modulus, UTS and maximum elongation from the stress vs strain curves.
This is the conclusion from the observation of experimental phenomena.
- Additionally, i have soume doubts about the methods adopted. Were teh measurments repeated? In any case you must provide data with the related uncertainty values.
We repeated it many times to ensure the accuracy of the data.
- Furthermore, i suggest to add a section where the performances of teh material herein reported is compared with literature data. This will add a significant information to the manuscript.
In order to further highlight the excellent electromagnetic absorption performance of CNTs@Fe3O4/PU, CNTs@GN@Fe3O4/PU and multilayer magnetic foam prepared in this article, the electromagnetic absorption performance of various Fe@carbon-based nanocomposites in recent years The corresponding values are listed in Table 5.1. It can be seen intuitively from the table that the overall performance of the magnetic foam prepared in this article is significantly better than other materials, no matter from the comparison of the filling amount, absorption strength, matching thickness or width of the absorption frequency band of the absorber.
Table 5.1 Statistics of electromagnetic absorption performance of Fe@carbon-based nanocomposites in recent years
|
Microwave absorber |
Filler loading wt% |
Thickness(mm) |
Frequency (GHz) |
RLmin value (dB) |
Bandwidth RL<−10dB (GHz) |
Ref. |
|
Fe3O4@C/CNTs |
30 |
2.0 |
9.66 |
-35.7 |
3.06 |
[22] |
|
Fe3O4/C/CNTs/GO/Paraffin |
20 |
1.9 |
11.4 |
-54.43 |
3.2 |
[110] |
|
CoFe2O4/MWCNTs/RGO/Paraffin |
20 |
1.6 |
11.6 |
-46.8 |
3.44 |
[111] |
|
Fe/Fe3C/MWCNT/ Paraffin |
30 |
2.0 |
13.9 |
-31.3 |
2.8 |
[112] |
|
Fe3O4/Ppy/CNTs/ Epoxy |
20 |
3.0 |
10.2 |
-25.9 |
4.5 |
[113] |
|
CNTs/GNS@ CoFe2O4 |
30 |
3.0 |
10.3 |
-29.1 |
3.47 |
[114] |
|
RGO/AC/Fe3O4 |
5 |
3.0 |
11.64 |
-22.24 |
5.53 |
[115] |
|
GN/Fe3O4/C |
25 |
1.8 |
14.8 |
-30.1 |
5.4 |
[116] |
|
MWCNTs/ZnFe2O |
50 |
1.5 |
13.4 |
-55.5 |
3.6 |
[117] |
|
70CNTs@Fe3O4/ PU |
15 |
2.0 |
13.17 |
-43.51 |
4.36 |
This work |
|
50C2G1 |
15 |
2.28 |
10.33 |
-59.44 |
6.09 |
This work |
|
D3 |
15 |
2.5 |
9.42 |
-42.86 |
5.98 |
This work |
|
S5 |
15 |
3.0 |
8.74 |
-65.38 |
5.28 |
This work |
Reviewer 3 Report
The paper presents a study of possibilities to absorb electromagnetic waves with a novel composite graphene-based foam supplemented with Fe304. The subject of the paper is within the scope of the journal Materials.
The paper is well written and it deserves publication "as it is". As correctly pointed out by the authors the problems related to electromagnetic "pollution" are of paramount importance in contemporary era of widespread use of electronics devices. Thus the issue raised in the paper is indeed very important from the practical point of view.
The authors have taken into account different applicational aspects of the considered six compounds differing in the ratio of @GN@ Fe 3 O 4 nanoparticles to polyether polyurethane %weight (in particular absorbing properties, quasi-static magnetic properties expressed with hysteresis loops, mechanical properties etc.) The paper should attract the attention of the scientific community and result in numerous citations.
I congratulate the authors on impressive and interesting work !
Author Response
we have improved the paper.
Reviewer 4 Report
In this study by C. Gao et al. titled Electronic absorption and mechanical properties of composite foams for CNTs@GN@Fe3O4/PU), carbon nanotubes (CNTs) @graphen (GN) @Fe3O4 / polyurethane(PU) nanocomposites were synthesized and were investigated for their electromagnetic absorption characteristics in frequencies ranging from 2 to 18 GHz. and mechanical properties. The authors have found material and thickness conditions for achieving a full coverage absorption of the X band frequency region. The electromagnetic absorption mechanisms were analyzed in terms of dielectric constant and magnetic permeability. Resonance features observed in dielectric constants were assigned with defect mediated polarization centers present in GN / CNTs interfaces, while resonance features observed in magnetic permeability were assigned in terms of usual magnetic resonances in Fe3O4. Based on these assignment, the authors states that the electromagnetic absorption by CNTs@GN @Fe3O4 / PU nanocomposites are caused by the synergistic effects of these dielectric and magnetic losses.
It follows from the following reasons that this manuscript is not recommended for publication in Materials.
First, to confirm the presence of synergistic effect of dielectric and magnetic losses (L. 335~L. 337), measurements on two types of samples are indispensable: (i) CNTs@GN / PU and (ii) Fe3O4 / PU because type (i) sample can give dielectric and permeability characteristics without influence from Fe3O4 and type (ii) sample can give those characteristics without influence from CNTs and GN. The sample studied are, however, CNTs@GN @Fe3O4 / PU nanocomposites, which cannot prove the presence of synergistic effects of dielectric and magnetic losses. Complemental sample preparation and measurement are essentially necessary for revealing the synergistic effects.
Second, regarding the authors’ proposal for electromagnetic wave absorption mechanisms, multiple scattering and reflection in GN sheets (L. 354~L. 355, L. 363~L. 365, L. 377~L. 380. and Figure 7) cannot be accepted, because the wavelength used in this study is approximately several cm, which are extremely larger than size of the elements consisting of CNTs@GN @Fe3O4 / PU nanocomposites. Comparable dimension between wavelength and the scatters of electromagnetic wave is necessary for the multiple scattering and reflection.
Third, regarding the authors’ interpretation for the resonance bands observed in the dielectric constant around 10 GHz, the authors assigned the resonance band with the defect mediated polarization centers (L. 349~L. 351, L. 358~L. 362). According to the authors’ interpretation, the polarizations are generated in C-O and C=O owing to electron affinity difference between C and O (L. 358~L. 362). This assignment implies that the resonance frequencies are several THz, because C-O stretching and bending modes are responsible for the resonance frequencies. The dielectric resonance was, however, observed around 10 GHz as shown in Figure 5(b) and Figure 6. An alternative explanation for the low frequency resonance mechanism is necessary.
Fourth, a number of improper expressions and usages were observed in this manuscript. They are for instance as follows.
- In L. 22, “PU” is used without its definition.
- “Nano” in L. 22 must be “nano”.
- In L. 25, “Finally” must be “Second”.
- In L. 27, “50c2g1” is used without its definition.
- In L. 27, “50c2g1” must be “50C2G2”, because the latter type is used later.
- In L. 27, “rlmin” is used without its definition.
- In L. 27, “rlmin” must be RLmin because the latter type is used later.
- In L. 40, “EMWA” is used without its definition.
- In L. 51, “Oh” must be “OH”.
- In L. 66, “CRT” is used without its definition.
- In L. 136, no name of production company for FE-SEM.
- In L. 137, no name of production company for X-ray spectrometer.
- In L. 141, “0.11542 nm” is incorrect value for Cu Kα lineIn.
- In L.147, “EC” is used without its definition.
- In L. 190~ 193, the same sentence is repeated.
- In L. 197, “Good” must be “good”.
- In L. 237, “With” must be “with”.
- In L. 244, “50C1G1” is used without definition.
- In L. 463, no volume and no page numbers for Reference 9.
- In L. 486, no volume and no page numbers for Reference 19.
- In L. 506, no Journal name for Reference 29.
- In Figure 1, character sizes are too small.
Therefore, this manuscript is too incomplete to review.
Author Response
we have improved the paper.
Round 2
Reviewer 1 Report
The revised manuscript still contains numerous faults and drawbacks.
- Abstract must be rewritten: lines 15 to 19 should be excluded; CNT and GN abbreviations must be explained; 'rlmin' - must be explained. And just read the Abstract carefully to realize what you want to say to readers.
- Text of the manuscript is very negligent in general and difficult to read. Style of presentation may be suitable for technical reporting but not for academic publication. The manuscript contains numerous statements about 'obvious' character of conclusions. If it is really 'obvious' the results of work are trivial and have no need to investigation and reporting. But, in fact, there are not obvious, of course, and require analysis which is practically absent in the manuscript.
For example, one of the main assumption proposed consist in existence of '...defects ... which will cause polarization loss...'. What kind of defects? How these defects are formed? How they are effected on EM absorption? etc.
- There are technical problems in results presentation:
- Fig. 2a presents plate like structures named in the text as 'graphene nanosheets'. The dimensions of these plates exceeds 200 um, but accordingly to Section 2.1 'Graphene nano-platelets (GNPs) has an average diameter of 10 μm'.
- Fig. 2b presents something which is mentioned as 'CNT' structures. But accordingly to Section 2.1 'Carbon nanotubes (CNTs) with 9 nm outer diameter and 1.5 m length'. Resolution of used SEM is below to the limit allowing observation of such kind of nanotubes
- there are no 'sandwich-like' structures on Fig. 2
- Style of Fig. 1 and Fig. 7 is similar to comic strip and does not provide any information different from those presented in the text
- D and G bands in the Raman spectra presented at Fig. 3d belongs to sp2 (graphitic) carbon; there are no bands which may be assigned to sp3 carbon. At the same time origination of bands in low-frequency range (below 750 1/cm) is not clear; what it is?
Reviewer 2 Report
Neither one of mine concerns have been answered. Mechanical proeperties from stress vs stress curves were not extrapolates. Uncertanty values were not reported. The answers provided from the authors are not sufficent to improve the quality of the manuscript up to the mark for pubblication.
Furthermore, thay affirmed that they repeted the measurments many times but not error values are provided. Numbers without any uncertanty value are just a bunch of numbers and not science.
At the present stage considering the insufficent efforts of the review work, i cannot endorse the pubblication of this paper. Accordingly, i recoomend to reject it.
Reviewer 4 Report
This revised manuscript is not recommended for publication in Materials, because (i) no response letter for previous review report was prepared by the authors , and (ii) no descriptions responding to previous reviewer's comments was observed in this revised manuscript.